# Effect of Artificial Regime Shifts and Biotic Factors on the Intensity of Foraging of Planktivorous Fish

**DOI:** 10.3390/ani12010017

**Published:** 2021-12-22

**Authors:** Krzysztof Ciszewski, Wawrzyniec Wawrzyniak, Przemysław Czerniejewski

**Affiliations:** Department of Fisheries Management, Faculty of Food Sciences and Fisheries, West Pomeranian University of Technolgy, Królewicza 4, 71-550 Szczecin, Poland; wwawrzyniak@zut.edu.pl (W.W.); przemyslaw.czerniejewski@zut.edu.pl (P.C.)

**Keywords:** climate change, planktonic animals, foraging fish, regime shifts

## Abstract

**Simple Summary:**

We studied the effect of elevated temperatures by creating an artificial stratification of water resembling a biotope, e.g., a lake, including natural biotic factors in the water column. Elevated temperatures caused changes in biotic factors and abiotic gradients. As a result of these changes, the foraging of planktivorous fish on small-sized zooplankton increased. The small size of zooplankton resulting from the changes in gradients was mainly due to the improved adaptation of small organisms to decreased oxygen and food availability. The smaller size also decreased the visibility of the prey to the predator.

**Abstract:**

It is still to be confirmed whether global warming with its predicted elevated water temperature will cause an increase in predation and alter phenological and physiological processes leading to changes in the size of aquatic organisms. In an experimental system of water column stratification simulating a natural combination of field conditions, we created artificial abiotic factors that mimicked the natural environment, i.e., light intensity, oxygen conditions, and thermal stratification. Subsequently, we added biotic factors such as algae, *Daphnia*, and planktivorous fish. We studied the intensity of foraging of planktivorous fish on individuals of *Daphnia* per min in different conditions of biotic and abiotic gradients. We demonstrated a possible scenario involving the risk of elimination of large prey within macrocladocera communities by predatory pressure as a result of climate change. A higher intensity of foraging of planktivorous fish caused or increased the occurrence of larger groups of planktonic animals with a smaller body size. The mechanisms of a future scenario were discovered at a higher trophic level in the aquatic environment.

## 1. Introduction

Macrocladocera of the *Daphnia* genus are considered a model aquatic organism for biologists, globally. The vulnerable *Daphnia* play a particular role in the diagnostics of changes in the aquatic environment [1,2].

Elevated water temperature causes an increased predation rate of planktivorous fish. Moreover, *Daphnia* respond to changes in temperature/thermocline and oxygen/oxycline. *Daphnia* prefer high temperatures and produce hemoglobin in response to oxygen shortage, which promotes oxygen distribution to all tissues in these high temperatures.

The regime shift effect shapes abiotic gradients, i.e., thermocline and oxycline, and biotic factor shifts, i.e., food availability [3]. The observed changes, and most of all climate warming, cause a decrease in oxygen levels in surface waters, boosted by an increased decomposition of matter and by changes in thermal stratification.

A study of a dozen lakes in the temperate climate zone confirmed decreased oxygen levels which are common in surface habitats and deeper layers and could threaten essential lake ecosystem processes [3]. Moreover, multiple studies have confirmed the influence of spells of hot weather on the thermocline change from gentle to sharp, with the resulting oxycline change in lakes [3,4]. Smaller-capacity zones with good oxygen conditions are crucial in maintaining competitiveness and diversification of all aquatic animals. It can be hypothesized that the increased prevalence of small-bodied species in zooplankton communities and the enhanced swimming speed of fish, especially planktivorous fish species [5,6,7], eventually cause changes in the intensity of foraging of planktivorous fish.

The literature regarding the effect of zooplankton population density on its distribution in predation risk gradients is less consistent. Some studies revealed that planktonic animals, such as the *Daphniidea* family of Cladocera, create tighter aggregations in the presence of predation risk to visually oriented planktivorous fish [8,9] and that fish are more confused when they encounter prey aggregations [9]. On the other hand, other studies suggested that aggregating may be maladaptive as an antipredation defense, as fish increase their foraging activity in patches of prey [10,11,12,13]. It is expected that high-density *Daphnia* in deep water will have a reduced the risk of mortality [14]. Larsson and Lampert [15] showed that deep water not only reduced the risk of mortality, but also acted as a low-oxygen refugium with available food, i.e., bacterioplankton [16]. Diel vertical migration (DVM) and depth distribution of planktonic animals have been reported in many studies [2,14,15,17,18,19,20,21,22,23,24,25,26,27,28,29]. Nevertheless, the vertical distribution of large daphnids, such as *Daphnia*, in the water column of a lake is generally uneven [2] due to the limitations of individuals as well as the structuring of *Daphnia* communities and the segregation of the genotype in the vertical profile [30,31]. *Daphnia* most often have to optimize their location [11]. Although several authors focused on the size distribution in the plankton tower, usually observing diel migration of plankton, similarly as in our water columns [14,15,20,24,25,32,33,34,35,36], to our knowledge only one study showed size changes in the distribution of a *Daphnia* population after fish introduction into a plankton tower [14]. The literature does not provide any examples in which the effect of population density were tested in the overlapping gradients of food concentration and predation risk, with a trade-off in the maximization of density-dependent net food and the minimization of density-dependent predation risk in the quest of every individual to find the most profitable location. To our knowledge, the intensity of fish foraging in relation to *Daphnia* density at gradients of temperature and oxygen—new conditions corresponding to climate change—has never been shown before.

Swimming behavior and depth distribution of planktonic animals are important components of zooplankton ecology [20,37]. Multiple authors [19,37,38,39,40] identified a few major areas in individual swimming behavior as important mechanisms. Nevertheless, the swimming behavior is generally an individual and species-specific mechanism [2,28]. The strongest factor for DVM in macrocladocera is light intensity [41,42] or risk of predation in general. Although light is considered the major proximate factor, including the antipredator response of *Daphnia* against visually oriented predators, some studies revealed that the reaction of *Daphnia* to chemical cues produced by predator strengthens the reaction to light [43]. Nevertheless, during migration, *Daphnia* incur high physiological costs (eventually resulting in escape or trade-off) [23,43,44]. Animal ectoderm, including that of *Daphnia,* may bear the physiological costs of vertical movement, for example, the production of heat shock protein (HSP70) [44,45,46]. Change in oxygen concentrations from 8 mg O_2_ L^−1^ to 3 mg O_2_ L^−1^ generates an expense of hemoglobin (Hb) production. Nevertheless, variations between individuals and differences between species and clones exist in *Daphnia* [32,39], e.g., the *D. longispina* population of lake El Tobar in Spain is known to be clonally diverse, and changing genotypes could play an important part in the observed seasonal differences in behavior and size [26,46]. Moreover, although *Daphnia* species bear physiological costs during temperature changes, the LOR1 clone is physiologically capable of producing Hb and HSP.

Additionally, elevated temperature and low oxygen level in the aquatic environment are very important because of shrinking zones with available food concentrations and increasing areas of poor food and lower energy availability. In the natural environment in moderate climate, *D. longispina* occurs in lakes from the end of May until July [16,26,47], but if the top–down pressure is low, the species occurs later in the season [47]. Although the total level of species distribution changes, the phenomena occurring up-scale from individual species to ecosystems are not elucidated by the current scientific knowledge.

In the present study, we tested three hypotheses.

**Hypothesis 1** **(H1).**
*Elevated temperature causes a change in the thermal gradient from gentle (L) to sharp (O) and in the oxygen gradient from normoxia (N) to hypoxia (H), as well as in plankton distribution, which increases the foraging by planktivorous fish;*


**Hypothesis 2** **(H2).**
*Elevated temperature and the consequent thermal gradient change cause a higher selection foraging rate in planktivorous fish with respect to prey of greater size;*


**Hypothesis 3** **(H3).**
*Higher temperature causes a higher selection foraging of planktivorous fish with respect to prey with elevated hemoglobin (Hb) level.*


## 2. Materials and Methods

### 2.1. Mesocosms–Water Columns

We used two indoor mesocosms, up to 200 L each. This was a small-scale experiment that mimicked a natural combination of field conditions (Figure 1). We used two water stratification columns previously described [10,43], with modifications. The water columns with kairomone were equipped with a flow system allowing a fish density of 0.01 fish L^−1^. First, we introduced algae to the columns stratified for thermal and oxygen conditions. The introduced algae were then gently mixed. The concentration of phytoplankton in each sample taken from subsequent depths was assessed by measuring chlorophyll concentration using a fluorometer (Turner Desing, San Jose, CA USA). After algae stratification, we used jars to introduce *Daphnia* to the columns at a density of 2 individuals per L, and after two hours, we used nets to introduce acclimated planktivorous fish and started the experiment. Each experiment lasted 10 min. We used new empty columns (equipped with a two-drain valve and counting the remaining *Daphnia*) in all experiments, as shown in Figure 1. The incubation of fish in preparation columns before their introduction lasted approximately 12 h. The experiments were conducted in 2017, in series and in cycles conducted every other day (for a schematic of the experiments, see Figure 2 and Appendix A). For each tested Hypothesis (1, 2, and 3), separate controls were used.

### 2.2. Preparing Light, Oxygen, and Thermal Conditions

Cooling and heating water in the columns for thermal stratification was immediately provided by an EK30 immersion cooler (Thermo Scientific Ltd., Waltham, MA, USA) and an ADA-REX ZEFIR water heater with a thermoregulatory capability. Stratification (thermocline) was generated with the gentle and the sharp gradients. The thermal gradients L and O are conventionally shaped thermoclines: L, very gentle temperature decrease in the water column and O, locally very sharp temperature decrease near thermocline in the water column [48,49]. Measurements of temperature and oxygen concentration were conducted using a ProODO optical oxygen meter (YSI). An Aqua Lifter AQ-20 Dosing Pump was used to aerate the water column according to the expected oxycline H hypoxia event concentration ≤0.2 mg O_2_ L^−1^ and N normoxia concentration of 5.5–7.5 mg O_2_ L^−1^. Hypoxia conditions were created by pumping 100% purified nitrogen. Light intensity mimicked natural evening underwater light during intense planktivorous foraging. Artificial light intensity was obtained via special filtration using a frosted glass screen proposed by Maszczyk [20], with a 60 W halogen light as the only light source. We obtained a surface light intensity of approximately 2.5 µmol m^−2^ s^−1^, measured based on the photon flux density using a LI-COR LI-189 photometer quantum sensor (Lincoln, NE, USA).

### 2.3. Daphnia

*D. longispina* clone LOR1 (derived from caught *Daphnia* that became the progenitor of the cohort) was selected because of its pronounced response to light compared with other clones, and light intensity in the column was an important parameter in our study. We used *D. longispina* clone LOR1 from lake Roś, Great Masurian Lakes, Poland, as it is a common and typical planktonic animal in the pelagial of European lakes. The LOR1 clone was cultured in laboratory batch cultures in 4 L glass jars with *Chlamydomonas reinhardtti* (1 mg C_org_ L^−1^ for 2 days) with a 12:12 L/D regimen. Water for cultures with *D. longispina* clone LOR1 was filtered through a 1 µm filter. In the experiments, we used only 5-day-old clones with a body length of approx. 0.992 ± 0.129 mm. Our earlier study showed than this clone did not react to kairomone, or the reaction was not clear, but showed a strong reaction to light intensity and low oxygen levels. Before the experiment, the clone was acclimated to light and dark glass jars. The hemoglobin content of *D. longispina* LOR1 individuals was determined using a modified method described in the literature [50]. To obtain *D. longispina* density data and fish images during the experiments, we used a special underwater video camera—Type KSPP-25/700 transformed SONY 1/3” 700 TVL, HD lens 3.6 mm F 1.4, chassis IP 68, Akrylon XT (Matras Ltd., Warsaw, Poland)—with automatic motion. In one variant, we used a *Daphnia pulex* clone from Lake Roś, Great Masurian Lakes, Poland, with a body length of 1.73 ± 0.05 mm, During the experiment, the density of the *D. longispina* clone LOR1 decreased to values which allowed the presumption that the natural conditions were well simulated. Changes in species distribution in space and depth result in a new ecosystem composition, the elimination of some species, and changes in the trophic level; however, scaling from individual species up to ecosystems currently constitutes a gap in the scientific knowledge. Capture rate was determined based on the quantity of *Daphnia* individuals consumed divided by two fish individuals in the column and further divided by the duration of the experiment (10 min). This yielded the *Daphnia* capture rate per individual fish × min^−1^.

### 2.4. Fish

The fish rudd (*Scardinius erthophthalmus)* was reared in a hatchery at the Pond Fishery Department (Żabieniec, Poland) and transported to the experimental site (Department of Hydrobiology, University of Warsaw); it reached a mean body weight of 2.55 ± 0.38 g and a mean body length of 113.4 ± 0.23 mm and was acclimated at a temperature of 21 °C before the experiments. LOR1 and ruddy were not fed one day before the experiments. Two individuals per column were used in the experiment, as this corresponded to the fish density found in eutrophic lakes. After treatment, fish were transferred to the second column. Fish were exchanged two times in a series.

### 2.5. Data Analysis

The Wilcoxon test was used because of the absence of normal distribution in the K-S test of variables measured. This test is a non-parametric version of the Student’s *t*-test for independent samples. Z-scale results were tested for statistical significance. Z-scale is the Mann–Whitney U test statistic, just as T is a measure of Student’s *t*-test. The significance of Z-scale differences can be established by comparison with tables. However, we actually calculated *p* < 0.05, supporting our conclusion that the two compared samples differed in a statistically significant manner. The calculated *p* < 0.05 denoted a statistically significant difference. Alfa = 0.05 was adopted because the investigated sample size was <1000. Statistical analyses were conducted using STATISTICA 13.3 software from StatSoft Poland [50,51,52,53].

## 3. Results

The capture rate was higher in a statistically significant (the difference was statistically significant at the level of *p* < 0.05) manner compared with that of controls under hypoxia (HL and HO), but did not differ significantly under normoxia (NO) (Hypothesis 1, Table 1 and Table 2). This was due to the fact that hypoxia causes greater foraging of planktivorous fish, as opposed to normoxia. The shape of thermocline had a small effect on foraging. In A2, the *D. pulex* capture rate in planktivorous fish in NO was higher in a statistically significant manner than the *D. pulex* capture rate in the control group in A2, as shown in Table 1 and Figure 3. The capture rate of *D. pulex* in NO was higher in a statistically significant manner than that of *D. longispina* of the control group in A2, as shown in Table 2 and Figure 3. The greater foraging was caused by the larger dimensions of the prey and the sharp shape of the O thermocline. In A3, under hypoxia and O and L thermoclines, the capture rate was higher in a statistically significant manner in fish foraging on *D. longispina* with increased Hb level in A3, as shown in Table 2 and Figure 3.

## 4. Discussion

The results of the simple experiments described here proved and generally confirmed our three Hypotheses: 1, claiming that an elevated temperature causes an oxygen gradient change from N to H and a thermal gradient change from L to O, as well as increased foraging of planktivorous fish; H2, claiming that an elevated temperature causes a greater selective foraging rate of planktivorous fish with respect to larger prey; H3, claiming that an elevated temperature causes greater selective foraging of planktivorous fish with respect to their prey with elevated Hb content. The increased foraging rate of planktivorous fish was affected by temperature and *Daphnia* distribution but not by the volume of the zone mimicking epilimnion. This observation was confirmed by the length of dwelling of rudd in the given zone and by *Daphnia* distribution (see Appendix A).

Our study with *D. longispina* and *D. pulex* species A2 focused on body size variations suggested that there are individuals of different body size in the environment, with increased pressure on the larger ones, as shown in Table 2 and Figure 3. A high fish predation pressure is an important factor controlling the size structure of a macrocladocera population, including *Daphnia*; the presence of predators induces a morphological shift in body size [35,53,54,55]. A study by another group [53] showed that *Daphnia*, depending on fish predation pressure, could decrease their body size by 50%. This result was supported by another study [56] which compared the body size of *Daphnia* from lakes with different fish predation pressures. Another study showed the effect of critical conditions on the near thermocline, i.e., poor food availability, decrease in oxygen concentration, as well as avoiding light and altered hemoglobin production by *D. longispina* in Mekkojärvi lake in southern Finland. A decrease in the body size of *D. longispina* could be caused by predator pressure, e.g., the invertebrate *Chaoborus* [47]. Moreover, even when the predator is separated from its prey, the body size decreases, and it is suggested that there is some chemical signal inducing these morphological adaptations [57]. Certain *Daphnia* species have elongated spines, helmets, or neck spines as defense against predation [58]. Such defenses are energetically costly, and a smaller body size may be beneficial, because energy can be redirected from locomotion to reproduction [58,59]. A smaller body size of planktonic animals is promoted by a high predation pressure (including lower prey visibility by predators) and a lower physiological expense of energy (including lower use of oxygen, beneficial when oxygen availability in the aquatic environment decreases). In the future, one can expect that elevated temperatures will result in an increase in the proportion of planktonic animals with a small body size in zooplankton communities. In our study, we used *Daphnia* clone LOR1 with a significantly higher fish predation pressure at elevated temperature and a zone epilimnic effect, as well as a high rudd capture rate. Moreover, the depth distribution of *D. longispina* LOR1 was common for lakes during the day and with high light intensity. A study [34] reported that *D. longispina* remained rather close to the surface near the thermocline during both the day and the night. Another study [26] confirmed this result when analyzing the diel distribution of the natural population of *D. longispina* in the vertical profile in El Tobar lake in Spain; these authors reported that *D. longispina* occurrence ended at a depth of 9–12 m of the thermocline. Nevertheless, adults dwelled predominantly in rich, deep waters near the oxycline, which suggests that adults are more resistant to lower oxygen levels and lower temperatures [26]. The latter study, in accordance with our results, showed that *D. longispina* had a similar depth distribution in the evening with the same light intensity and other natural conditions. This can indicate that the presence of *D. longispina* close to the surface is the same in different climates [26]. The distribution of *D. longispina* in Mekkojärvi lake in another study [47] was similar to the one we observed and indicated that *D. longispina* from the Finnish lake had no synchronized migration; their bimodal vertical distribution rather suggested an asynchronous vertical migration. Nevertheless, some individuals showed a particular tendency to concentrate near the oxycline, close to the dense phytoplankton and bacteria populations in the upper part of the anoxic hypolimnion. Usually, when *D. longispina* is young (3–4 days old), it stays in lakes near the surface both at night and during the day, but when it is old, it migrates between zones near the surface at night and near the thermocline at day. Nevertheless, the *D. longispina* clone LOR1 both acclimated and non-acclimated to lower oxygen levels avoided zones with lower oxygen concentration. A study [60] showed that the macrocladocera size may be reduced under low food conditions, with increased time to maturity. Another study [61] also showed that the age of first reproduction was delayed when food was limited. Brancelj and Blejec [62] found large sizes in September, with a strong tendency to reach deeper levels, which protected them against visual predation of the young. Body speed (BS, cm × s^−1^) is an important factor of foraging planktivorous fish [63] because a high BS causes higher ruddy foraging. Another study showed than high temperatures from 16 °C to 26 °C resulted in a high BS, from 5.07 cm × s^−1^ at 16 °C to 8.51 cm × s^−1^ at 26 °C [63], and in a high encounter rate, from approx. 3 prey × min^−1^ × fish to 7.38 prey × min^−1^ × fish. In experiments with different *Daphnia* densities, the mean body length of *Artemia* sp. was 0.4 mm, while the speed of fish during foraging at temperatures of 20–23 °C averaged 5.62 ± 1.99 cm × s^−1^ [11]. In our study, the mean body speed during the experiments was 5.35 ± 0.08 cm × s^−1^. The body speed of rudd was similar to that reported in many other studies, as well as that recorded in the natural environment. Artificial light used in our study was similar to natural light, as we considered the fact that it may cause prey mortality due to visually oriented predation [42]. Other studies usually did not use light similar to the natural light, for example, ~350 µmol quanta m^−2^ s^−2^ measured at the surface [15], which disturbed the effect of foraging fish. However, the temperature and BS were quite similar. Some studies also showed a high BS after a decrease in prey density [5].

In the near future, in the European climate zones, there will be a high number of eumyctic lake types in the summer, with elevated epilimnion and metalimnion temperatures at deeper levels without water circulation, as well as lower water exchange during the spring and autumnal circulations owing to warmer winters, which in turn will result in adverse conditions for bacteria, phyto- and zooplankton, and fish. In most lakes, including warm monomictic lakes of central Europe, the oxygen conditions are deteriorating [3,64]. For organisms using oxygen, this causes environmental stress. Such phenomena may not always be readily noticeable.

Therefore, we tested different conditions, simulating first those before warming of the aquatic environment and then those corresponding to the global warming, and concluded that the new habitat requires a better physiological preparation of planktonic animals [3]. Higher temperatures cause lower oxygen levels and higher fish foraging, resulting in the smaller size of planktonic animals [53,65]. The specific species not only have small bodies, but also show higher reproduction and better adaptation to lower oxygen concentration and higher pressure of planktivorous fish, as well as broader epilimnion areas. As a possible scenario, changes in stratification will occur, and a large portion of the aquatic environment will be epilimnic, with decreased oxygen concentrations and high temperatures [3]. These changes will result in an increased prevalence of small-bodied species in zooplankton communities, as well as in their higher specialization. In the new habitat, due to the climate change, it will be beneficial for macrocladocera to be smaller and better prepared physiologically for lower oxygen levels and temperature changes. In such a new habitat, higher water temperatures increase fish predation pressure and body speed, which selectively eliminates larger and more visible prey. The new habitat also causes changes at the trophic level. We defined a mechanism which global warming will introduce into the aquatic environment, predominantly including the warming of the upper zone in lakes and oceans following the elevated air temperature. In the warming aquatic environment, low oxygen solubility causes its lower availability, which in turn increases competition among all aquatic animals for oxygen. Although aquatic animals have a narrow scale of tolerable temperatures, a decrease in the available oxygen is much less tolerated than an increase in temperature in the new habitat, especially by communities of *Daphnia* with deep distribution. High temperatures will first accelerate the metabolism of planktivorous fish; subsequently, the feeding behavior will change to adapt to a higher body speed, aiming at effectively catching prey, with a simultaneously decreased body size. Fish show better adaptation to the metalimnic refugium compared with macrocladocera, as their predation pressure risk is reduced. The production of Hb, e.g., by *Daphnia*, helps respiration, but facilitates visualization by predators in these conditions. The effect of temperature, oxygen availability and high predator pressure will result in a domination of smaller species in zooplankton communities. Larger visible species will lose the competition to smaller and less visible ones. Moreover, species consumed more often will lose the potential for successful reproduction. A smaller body area will also require less oxygen. The greatest effect of climate change and elevated temperatures of the aquatic environment on macrocladocera communities will be the elimination of large prey and changes in their deep distribution, as well as an imbalance between mortality risk and reproduction in more challenging conditions.

## 5. Conclusions

Increased temperature causes changes in biotic factors and abiotic gradients. As a result of these changes, the foraging of planktivorous fish on small-sized zooplankton increases. The small size of zooplankton resulting from the changes in gradients is mainly due to the improved adaptation of small organisms to decreased oxygen and food availability. A smaller size also decreases the visibility of the prey to the predator.

## Figures and Tables

**Figure 1 animals-12-00017-f001:**
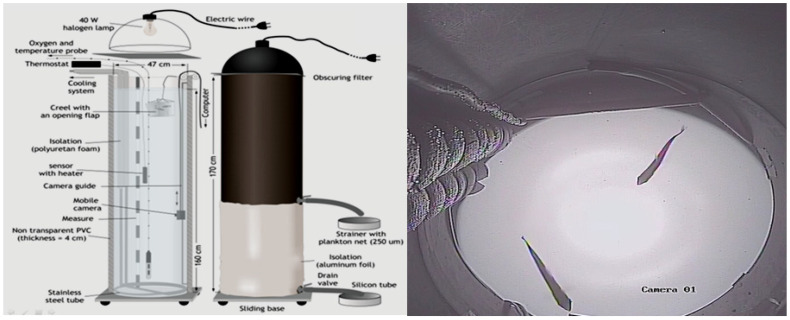
System of vertical columns used in the experiments and photos taken during the experiments.

**Figure 2 animals-12-00017-f002:**
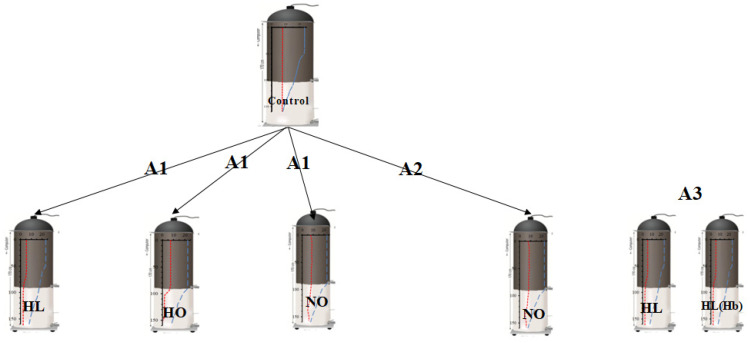
Diagram presenting the various experimental conditions.

**Figure 3 animals-12-00017-f003:**
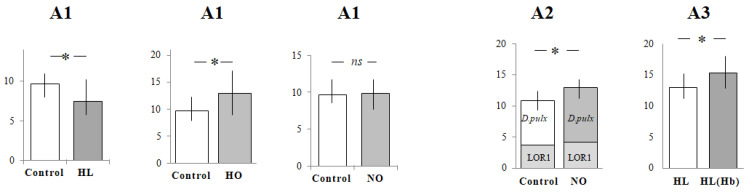
Wilcoxon test results for Hypotheses 1, 2, 3. * Statistically significant (*p* < 0.05); *ns* not significant.

**Table 1 animals-12-00017-t001:** Results of the experiments testing Hypotheses 1, 2, 3.

	(*n* = 4)			M (Mean)	Me (Median)	SD (Standard Deviation)	Min (Minimum)	Max (Maximum)
A1	Control			9.7	9.9	0.8	8.6	10.6
	HL			7.5	8.0	2.0	4.5	9.5
	HO			13.0	11.5	3.0	11.1	17.8
	NO			9.9	10.1	1.0	8.5	11.1
A2	Control	*D. pulex*	M	7.2	7.3	0.5	6.4	7.7
			11.0					
		*D. longispina*		3.8	3.9	0.4	3.3	4.2
	NO	*D. pulex*	12.9	8.8	8.8	1.3	7.3	10.3
		*D. longispina*		4.2	4.2	0.2	3.9	4.5
A3	Hb	HL (Hb)		12.4	12.2	2.4	9.4	15.6
		HO (Hb)		15.4	15.6	0.6	14.5	15.9

**Table 2 animals-12-00017-t002:** Results of the Wilcoxon test for Hypotheses 1, 2, 3.

		Pairs		Z	Wilcoxon *p* Value
A1	STATISTICA	Control and HL		−2.533	0.011
		Control and HO		−2.533	0.011
		Control and NO		−0.533	0.574
A2		Control *D. longispina* *D. pulex* NO *D. longispina* *D. pulex*	Control only ind. *D. logispina* and NO only ind. *D. pulx*	−2.533	0.011
			Control only ind. *D. pulx* and NO only ind. *D. logispina*	−2.533	0.011
			Control only ind. *D. logispina* and NO only ind. *D. logispina*	−0.558	0.011
			Control only ind. *D. pulex* and NO only ind. *D. pulex*	−2.533	0.011
A3		Control (HL *D. longispina*) and HL (*D. longispina* with Hb ↑)		−2.533	0.011
		Control (HO *D. longispina*) and HO (*D. longispina* with Hb ↑)		−2.111	0.035
		HL (*D. longispina* with Hb ↑) and HO (*D. longispina* with (Hb ↑)		−2.111	0.035

↑—high/elevated hemoglobin level.

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
