# Peer review of "Effect of Artificial Regime Shifts and Biotic Factors on the Intensity of Foraging of Planktivorous Fish"

_animals, 2021, doi:10.3390/ani12010017_

Round 1

Reviewer 1 Report

This manuscript presents the results of three mesocosm experiments conducted to determine the effects of oxygen level, zooplankton taxa, and hemoglobin content within zooplankton on the foraging rate of rudd (Scardinius erthophthalmus).  Below I provide comments on the manuscript that I hope will help the authors with improving the manuscript. 

Overall Comments:

A. line 17-18: I don’t see what you did as creating biotic gradients. You set up an experimental mesocosms containing algae, manipulated types and abundances of zooplankton, and then observed fish feeding rates on zooplankton. Thus, the experiments manipulated biotic factors of zooplankton type and abundances as well as abiotic factors of temperature and dissolved oxygen.  However, the experiments did not create a continuous gradient of zooplankton abundance or even a gradient in taxonomic composition.  Thus, I recommend revising title and content throughout entire manuscript to focus on manipulation of biotic and abiotic factors rather than biotic and abiotic gradients.

B. Introduction is a single paragraph and should be revised so that structure in multiple paragraphs

B. Methods needs more details and below I provide specific comments on what details are needed:

-need to clearly describe how the gentle L and sharp O temperature gradients differ with respect to water temperature and the range of values associated with each of these gradients

-need describe how hypoxia treatments and normoxia treatments differ with respect to the dissolved oxygen values and the range of values associated with each treatment

-The methods indicate that the authors used two mesocosms and that each mesocosm contained two water columns.  It is not clear but based on this I assume that the authors got 2 samples from each mesocosm during each of the five mesocosm trials.  Is this correct? 

-Figure 2 and information from the methods text suggest that the authors conducted a series of five mecosom trials with: 1) the control and HL; 2) the control and HO; 3) the control and NO; 4) the control and NO and Dpulex /LOR1; and 5) HL and HL(Hb).  Is this correct?  If so need to clarify this in the text of the methods and to report the dates that each pair of the five mesocosm trial was conducted and the sample size for the control and the experimental treatment in each of the five mesocosm trials.  Presumably each mesocosm had to be cleaned and set back up before running each trial, which suggests the experiments could not be done at once but on different dates.  If dates differ greatly then that might influence the comparison and as such this clarification is needed to enable the readers to interpret the methods.   Also, what is important here is that the authors need to confirm that for the four series of mesocosm trials involving the control treatment that a new control treatment was set up each time for comparisons with HL, HO, NO, and NO/duplex,LOR1.   If this is not the case then the authors need to explain what they did and their justification for it.  

-in the methods section need a separate paragraph devoted to describing the statistical analysis used.  Figure 2 implies the Wilcoxon test was used to compare if fish feeding rates differed between: 1) control and HL; 2) control and HO; 3) control and NO; 4) control and NO and Duplex/LOR1; and 5) HL and HL(Hb).  However, this needs to be specified in the methods section.  Additionally, need to state why used Wilcoxon test instead of t test, what statistics program was used to conduct the Wilcoxon test, and if the Wilcoxon test was used because the daphnia capture rate was not normal or have equal variance, then information on the tests used to confirm the lack of normality and equal variance needs to be provided. 

Specific Comments:

line 2-3: delete “in water columns”

line 5:  management is misspelled

line 7:  replace “study” with “studied” because your experiments were conducted in the past and as such past tense is needed here

line 7: delete “global warming”.  I concur your experiment has implications for understanding the potential effects of global warming, but you did not experimentally implement global warming as part of your experiment

line 7-9:  these two sentences are poorly written.  Revise and write one sentence that clearly summarizes your experiments

line 14: insert “predicted” between “its” and “elevated”

line 15: replace “causes” with “will cause”.  Need use future tense because this is discussing environmental conditions predicted to occur in the future

line 19:  delete “(hypoxia H)” and “(sharp O)”

line 31-32:  need citations to support this statement about how increased temperature affects predation rate of plantivorous fishes

line 32-33:  how do Daphnia respond to temperature changes and oxygen changes?  Need summarize for readers

line 33: which regime shifts will shape abiotic gradients?

line 37:  moderate zone of what and where?  Need be specific

line 39:  multiple studies conducted where and in what habitat types (streams, lakes, wetlands, etc.)?  

line 46: population density of what? 

line 61-62: what is a plankton tower? 

lines 104 to 461: due to the large number of places that I identified needing editorial comments I did not list them all here because that is beyond the scope of what I can do in this review.  My comments in the remaining part of the review highlight the major broad issues. I hope my suggestions for the Introduction section provide the authors with ideas on where the can improve and provide additional detail in the other sections of the paper. 

lines 104 to 158:  the methods section is one paragraph.  This is too long and needs to be restructured so its presented in multiple paragraphs to improve readability.

lines 159-166:  I don’t think this warrants its own subsection. 

lines 168-169: this is an interpretation of your results and it does not belong in the Results section but should be moved and discussed in the Discussion section.

lines 169-179: need to revise this to reference both the Figure 2 and Table 1better so that the reader can make the link between the text and the supporting information in the figure and table

lines 182-195:  I suggest splitting Figure 2 into 2 Figures with the top part (lines 182-190) being used in one figure to highlight the five mesocosm trials and the comparisons being made in each trial.  then the five subfigures with the results on lines 191-195 should be a third figure and used to convey the results. 

Table 1 is difficult to read and interpret.  What does M, Me, SD, Min, Max stand for?  what are the pairs that are associated with the Z and Wilcoxon p values – these were really hard to determine.  I suggest splitting this into two tables with one table being the first six columns and then the second table being the columns 8 to 11.  this will give you more space to work with the for the second table and enable you to make the table more easily readable for the readers. 

Reviewer 2 Report

The manuscript deals with a hot topic of climate warming and its potential effect on freshwater food webs. Though the authors formulated sound hypotheses, neither experimental design nor results are clearly presented in the current submission; hence, one can barely follow their arguments and cannot buy their conclusions.  Some parts of the manuscript are difficult for readers, in particular due to poor language, insufficient punctuation, numerous typing errors, etc. (see few examples in my specific comments below). English is awkward and the text deserves thorough definition of terms and their consistent usage, as well as language revision by a native speaker. Indeed as the manuscript deals entirely with large daphnids, i.e. Daphnia spp., the authors should use only these terms in a concise way, instead of confusing terms, such as Daphniidae, macrocladocera, etc. Moreover, both figures and tables are far of standard, self-explanatory requirements, and lack any legends or explanations of variables, so that it is really difficult to judge the results. To my opinion, the current submission should be immediately rejected by an editor and never been sent for peer-review, not even typeset.

Specific comments:

L. 7-13: Simple Summary is meaningless; English is awkward and deserves thorough rewording.

L. 19: the thermal… what? – missing word/definition.

L. 20: the family name Daphniidae should not be in italics throughout the text.

L. 23: fish gave rise to… – reword.

L. 31-32: Why should elevated water temperature cause an increased predation rate? – provide reference.

L. 34: food or energy? Here, and elsewhere throughout the text, the authors should clarify/define what they mean and carefully distinguish between food quantity and food quality, which largely matters in Daphnia growth and reproduction.

L. 48: Daphniidae is a family of the order Cladocera – correct.

L. 57: replace “macrocladocera” with “large daphnids” or “large Daphnia species” (or simply “Daphnia”) throughout the whole text; don’t use Daphniidae in this confusing way.

L. 92-93: This statement is not true; D. longispina often occurs later in the season if a top-down pressure is low, e.g. in [49].

L. 97: What does mean gentle (L) versus sharp (O)? Either thermal gradient deserves clear description or definition here; moreover, one should consider its distinct resistance (stability) in small experimental tanks compared to natural morphology of freshwater bodies. The authors must be cautious with over interpreting their results.

L. 105: Large or small-scale – provide the volume of the used mesocosms.

L. 112/123: check punctuation…

L. 118: Maszczyk? – provide a reference, yet not to an unpublished work [47] as elsewhere.

L. 138: The reference [47] is not “literature” = omit/replace it.

L. 140-142: Replace “organic carbon” with neutral “phytoplankton” or “food” and provide information about the food source for daphnids.

L. 146-149: Cross-check the given values for apparent miscalculation: if the mesocosm volume was ca. 0.28 m^3 (my guess from Figure 1), in fact, the two stocked rudds would have to correspond to approximately 8 individuals per cubic meter.

L. 151-156: How were both prey (daphnids) and predator (rudds) kept or acclimated before each experiment? Were fish hungry or fed until stocked into a mesocosm?

L. 160-166: Describe how/where daphnids and rudds were stocked – into the “epilimnion” or throughout the whole mesocosm? How were fish acclimated?

L. 170-171: Define both hypoxia and “normoxia”.

L. 172: As above, define clearly the shape(s) of thermocline (see also my comment on Figure 2 below).

L. 175-179: Unclear – what do you compare here to which control?

L. 182-196: Split Figure 2 (its caption is unjust as it holds only for the plots) to two separate items – the scheme of experimental setup and the results of statistics. Was indeed the volume of control the same as those of A1–A3 treatments? Redesign the scheme(s) and plots correspondingly for each hypothesis tested. In addition, provide full legends for all used items (treatments, species, etc.): e.g. red and blue lines (including readable axis scales!), white and light/dark grey column fills, etc.

L. 197: Provide self-explanatory caption of Table 1, with units and definition of all variables (M?,Me?).

L. 199-201: Was it really an effect of elevated temperature, or rather an effect of smaller space? Did the experimental fish stay in the “epilimnion”, or move within the whole water column?

L. 202: There is no proof of selection.

L. 215: Mekkojärvi – correct misspelling throughout the text.

L. 225: correct to “one can”

L. 227-228: What does mean “smaller clone” here – smaller to what? To my experience, Daphnia longispina can remarkably change its size under fish predation.

L. 228-229: What is “a high zone epilimnic effect”? – rephrase.

L. 269-273: Explain, how the climate change may result in adverse conditions? To my recent experience, there is a shift from dimictic to warm monomictic pattern of deep lakes in central Europe, yet nothing adverse compared to overall eutrophication, or fish overstocking…

Page 8: In general, the authors intended to discuss sexy hot topics of current ecology, yet they largely misinterpreted or over interpreted their results, if they used adequate experimental design.

L. 299: replace “specialization” with “adaptation”

L. 311: Again, conclusions are unjust and far beyond the experimental setup and results, as they are presented in the current submission under evaluation.

Pp. 9-11: Cross-check abundant typing errors in references, genus ad species names should be in italics, etc.

Reviewer 3 Report

Animals - Manuscript number 1448447

 Effect of Artificial Regime Shifts and Biotic Gradients in Water Columns on the Intensity of Foraging of Planktivorous Fish

General comments:

The manuscript has high scientific value highlighting experimental studies to evaluate increase of predation on larger zooplankton by planktivorous fish mimicked a natural combination of field conditions (abiotic and biotic).

The strong point of the manuscript is the robust methodological plan that resulted in valuable information about the effect elevated temperature i.e. global warming on interactions between trophic levels in aquatic environment under controlled conditions.

However, it is necessary to improve the material and methods section to organize in topics, to better explore the results obtained by applying some appropriate analyses. I highlighted some parts of the manuscript and inserted comments bellow.

Considering the aspects mentioned above, it is necessary corrections and improvements to make acceptable to Animals.

Specific comments:

 Introduction

The introduction is well written, but in a single paragraph. I recommend separating into separate paragraphs by topics covered.

Line 96: … “we tested”… instead “we decided to test”

Material and Methods

Materials and methods are very confused. It is necessary to describe in detail the experimental units (it is not enough just show Figure 1), the zooplaktonic organisms and planktivorous fish, as well as all the abiotic variables and the way in which these were evaluated (use of video camera, statistical analysis…). I recommend reorganizing by topic.

The authors should organize it in sections:

2.1 Mesocosms – water columns

2.2 Abiotic variables

2.2 Zooplankton – Daphnia

2.3 Planktivorous fish - Scardinius erthophthalmus

2.4 Data analysis

Line 12: insert dot after gradients

Line 118: … Maszczyk – author??or number of reference?

Lines 154-155: How was this evaluated? Was it evaluated by the video camera images, or analyzed the stomach contents of the fish? Authors must clarify these issues.

Lines 157-158: Clarify the statistical analysis used.

I recommend inserting images (photos of experimental tests) – fish predating Daphnia…

Line 165 - How many fish did you use in total? Was the same fish used in each experiment? Clarify it.

Results

Describe the results according to three hypotheses.

Line 168-179: Review all description of results according to the statistical analysis used. for example: ...a statistically significant manner... what does this mean ???

Table 1: numbers – dot instead comma

Discussion

Considering the results of the study and others already published that are mentioned in the discussion, I recommend elaborating a conceptual model of the effects of increased temperatures and other abiotic variables associated on predation pressure of planktivorous fish in relation to large zooplankton.

Round 2

Reviewer 2 Report

I'm pleased reading Authors' cover letter as they have addressed most of referee's comments. However, the revised text, still deserves thorough revision and editing - see some corrections and comments in the attached file. The final part of discussion is speculative and unjust. I strongly recommend its revision and reduction.

Author Response

Respons 

Reviewer 3 Report

The authors corrected and improved the manuscript according to my recommendations. 

Author Response

Respons 
